# Mehmet Ozan Sungurlu, the Legendary Turkish Petroleum Geologist

Oguz Mulayim[1]

[1]Turkish Petroleum Corporation, Adıyaman Directorate, Adıyaman, 02040, Türkiye

*Correspondence to*: Oguz Mulayim (oguzzmlym@gmail.com)

**Abstract**. On the occasion of October 29, 2023, the 100th anniversary of the Republic of Türkiye, I would like to pay tribute to M. Ozan Sungurlu (1939–1990), one of the most important non-academic competent Turkish field and petroleum geologists. and a pioneer of geological field observations throughout Türkiye. His scientific studies were published in numerous articles, many of which were co-authored by Turkish geologists. This article was written in memory and appreciation of M. Ozan Sungurlu, who made a great contribution to the field geology of Türkiye and whose importance in Türkiye is undisputed.

M.Ozan Sungurlu will be remembered as a valuable geoscientist whose passion and legacy we should share with future generations.

The occasion of October 29, 2023, the 100th anniversary of the Republic of Türkiye marks over three decades since the loss of M. Ozan Sungurlu (1939–1990), a famous representative of field geology. Attempting to illustrate his personality remains

a difficult endeavor as many of his contemporaries are deceased and no direct reports are available. In this context, only a professional biographer who has access to most of the necessary documents an often extremely rare situation could achieve a satisfactory result. Now. I find myself in a position where I hope to understand and express some of the personality traits and work of one of the leading personalities of field geology in Türkiye. Thirty-three years after his death, this article is not intended to be an exhaustive analysis of the life and work of M. Ozan Sungurlu but rather an overview of his career and contributions

to the geological studies of Türkiye. The "*Tribute to the Master*" by Balkaş (2014) and the recently published book *"My Brother Ozan Sungurlu"* by Dervişoğlu (2022) both provide a wealth of detail on the subject.

I would first like to pay tribute to M. Ozan Sungurlu, an outstanding field geologist whose name evokes deep gratitude in the hearts of his many friends and colleagues for the benefits he brought to both our country and the Turkish Petroleum Corporation (TPAO). The life and work of pioneer geologists such as M. Ozan Sungurlu represent a historical and fundamental reference

point that should be remembered by current and future generations of geoscientists. He passed down a rich legacy to all

geologists and bountifully shared his all-inclusive experiences, affecting them with his endless wonder. He was also a great intergenerational geologist, a motivational friend, and an excellent administrator.

M. Ozan Sungurlu was a Turkish geologist who was born in Gümüşhane, on October 20, 1939. His father Süleyman Faik Sungurlu was a government officer, teacher, and folk poet, while his mother Hatice Sungurlu was a housewife. Ozan was the youngest of seven children. After primary and secondary studies in Gümüşhane, the young man then followed the path of learning. Even though he was not one of the brightest students, he was good at the subjects that interested him (Dervişoğlu, 2022), and his teachers provided him with the valuable background knowledge that would form the basis of his later career.  As is common among youths of his age, those who choose a profession that follows their passion usually become diligent and smart students. Ozan attended high school in Eskişehir in 1956 and later graduated from İstanbul University with a diploma in geology and received a Bachelor of Science degree in 1964. There is sadly no known record of his geology studies at the University of İstanbul.  On December 22, 1965, he married his colleague Ms. Bilge from İstanbul University. Bilge earned a bachelor's degree in geology from Istanbul University. She was a devoted wife and life partner who always had Ozan's best interests at heart (Dervişoğlu, 2022). They have two children, Faik Sungurlu and Ebru Sungurlu.

M. Ozan Sungurlu began his career as a field geologist at the Mineral Research and Exploration Institute (MTA) between 1964 and 1967 and attended the exploration group of the TPAO in 1969. His extremely valuable contributions to the exploration effort led to his appointment as a vice manager of the Exploration Group in 1980, manager of the Exploration Group in 1984, and Vice President of TPAO in 1989. His dedicated, thorough geological studies throughout his life rightly earned him a reputation as an exemplary geologist and manager. He was President of the Turkish Association of Petroleum Geologists from 1980 to 1981.

His interest in integrated geological processes, where he became a researcher who used data and field relationships to challenge established dogmas for the geological problems in Türkiye. His extraordinary interest in the fundamentals of geological thinking drove him to solve the complex regional geology of Türkiye, especially the events from the Paleozoic to the present. He continued to be professionally active and published on the structural geology of southeastern Türkiye and the regional geology of the Turkish part of the Tethyan region with colleagues. He also contributed importantly to the geological studies of both TPAO and universities (e.g. mainly Istanbul Technical University and Istanbul University). M. Ozan Sungurlu's

credibility as a person of wholeness allowed him to be a bridge between TPAO and universities. Particularly, 1980-1990, released new fieldwork works to be produced in several studies that mentioned a new attribute of petroleum geology. For his colleagues and friends at Istanbul Technical University (İTU), Şengör et al. (1991) M. Ozan Sungurlu was: *"This extraordinary interest in earth sciences, combined with his superior intelligence and comprehension power, made him the greatest geologist*

*in Türkiye has ever been".* He earned the love and respect of all, not only through his achievements but also through his humble personality, as a sensitive, thoughtful, compassionate person, sincere and loyal friend, and his love of humanity. Most importantly, he was a truly caring person. He dedicated his time, energy, and skills to uphold the highest professional standards (Balkaş, 2014). His colleagues and friends found his sudden death unacceptable and are convinced that he will make many more contributions to his country and his profession.

M. Ozan Sungurlu emphasized the great importance of careful surveys and thorough analyses and repeatedly called on geologists to carry out field studies, especially those that precede the field, and to summarize all new data on the regional geology of petroleum areas in Türkiye (Balkaş, 2014). Even if he did not directly make the oil discovery, his colleagues in the Exploration group made many discoveries under his guidance. However, it is worth mentioning that his friends were amazed at the increase and diversity of his geological activity, his amazing working ability, and the abundance of his energy in the

petroleum areas of Türkiye. He had a unique style. Apart from TPAO company reports, Ozan Sungurlu had few publications in the literature. However, he has helped and mentored a number of his geology colleagues, including Prof. Dr. Doğan Perinçek, Prof. Dr. Yücel Yılmaz, and Prof. Dr. Celal Şengör. He is the best person to understand Türkiye's geology (Prof. Dr. Dogan Perinçek personal communication, February 2, 2024)) His slowness in composing articles is one of his weaknesses. Ozan Sungurlu is best described as someone who sees things others are blind to. The quote that best sums up Albert Szent-Györgyi

is: *"Research is to see what everyone else has seen and to think what nobody else has thought!"*

His studies interacted strongly with the best Turkish geologists, who are represented by men such as İhsan Ketin, A.M.C. Şengör, Yücel Yılmaz, Naci Görür, Fuat Şaroğlu, and Necdet Özgül. He tirelessly spent many years engaged in field research, through which he gained extensive mapping experience and mastered the analysis and synthesis of geological information. In the field, Ozan liked to draw, but he did not like to write very much. His contributions to field geology were so great that if he

were a painter, his name would be as famous as that of 'Picasso' (Prof. Dr. Dogan Perinçek personal communication, February

2, 2024). The enormous efforts he undertook and the skills to which he contributed have borne fruit worthy of him. Based on his comprehensive knowledge of Türkiye's regional geology, M. Ozan Sungurlu, through his brilliant problem-solving skills in field geology, effectively predicted new petroleum prospects and later realized these predictions through exploration. TPAO Report No. 871, authored by Ozan and written in Turkish, is the most famous report of its period and the most cited. The research included a thorough geological analysis of the region to determine which area is most favorable for oil exploration and how deep the Mardin Group (source and reservoir) lies beneath the Koçali and Karadut Complexes (allochthonous) and younger strata. The main focus of the research region is the structural belt of the foothills. Important tectonic events took place in this zone during the Upper Cretaceous. This geological study indicates that the foothills structure belt below the promontory is one of the possible oil targets. Thanks to this report, the structural elements and stratigraphy of the Adıyaman and Diyarbakır territories were finally understood (Balkaş, 2014). He impressed everyone with his gentle charisma, the clarity of his presentation, and the thoroughness of his argumentation. His audiences were captivated by his rich and vivid presentations of every geological problem he studied in different parts of Türkiye. He had a meticulous memory and could elaborate on the geology of all parts of Türkiye in fascinating detail. He emphasized the need to more intensively study the general geological problems of our country, including regional geology, stratigraphy, tectonics, and especially paleogeography. He encouraged all academics to work intensively on field geology and to conduct extensive field studies as part of an integrated approach.

In 1991, M. Ozan Sungurlu received the 'TÜBİTAK Service Award' from the Scientific and Technological Research Council of Türkiye (TÜBİTAK) for his work that significantly increased our country's oil and gas production through collaborative studies between the TPAO and universities.

As a sign of TPAO's loyalty to M. Ozan Sungurlu, in 1991, a well was named M. Ozan Sungurlu-1 after the oil discovery in the Karadut-1 well in Adıyaman, SE Türkiye. The field was subsequently developed and organized as the "*Ozan Sungurlu oilfield*". In 1991, his colleagues and the oil industry organized the *"Ozan Sungurlu Symposium Proceedings"* under the title *"Tectonics and Hydrocarbon Potential of Anatolia and Surrounding Regions"* and published scientific literature to honor his name and pass on his ideals to future generations. The valuable contributions of this symposium were summarized in a book that will be a lasting work for all geoscientists. In addition, a special issue of the Turkish Association of Petroleum Geologists (TAPG) Bulletin was published in 1992 in memory of M. Ozan Sungurlu (Balkaş, 2014).

The Ozan Sungurlu Foundation for Science, Education, and Aid was founded in 1991 by his colleagues to honor his name and has been operating for over 30 years with the support of his family. Since 1991, the foundation has awarded non-repayable educational scholarships to support financially disadvantaged geoscience students and the children of deceased TPAO employees in need of assistance. Throughout this periods, hundreds of students have been awarded scholarships. It is his wish to honor the memory of the late M. Ozan Sungurlu by ensuring that the service of the Foundation will continue for many years to come (Balkaş, 2014).

M. Ozan Sungurlu, vice president of TPAO, passed away on November 27, 1990, at the age of just 51, as a result of a tragic traffic accident. He was en route to his field studies and could not be saved despite all medical efforts. Almost 34 years after the death of this illustrious and extraordinary Turkish geologist, this humble compilation of his life and character intends to serve as a tribute, homage, and inspirational example to future generations of geoscientists in Türkiye and the rest of the world. His warmth, almsgiving, and enthusiasm for exploration were a source of enormous scientific inspiration for all those who worked at TPAO, and the whole TPAO community sorely misses his presence.

In 2001, thanks to the support and contributions of Prof. Dr. A.M.C. Şengör, the American Association of Petroleum Geologists (AAPG) elected to present an honorary award in the name of M. Ozan Sungurlu to a successful international student to be selected every year. M. Ozan Sungurlu was a shining example for many young geologists, and his legacy has allowed them to receive an international education and learn modern geological thinking overseas. In addition, he encouraged TPAO to participate in international meetings.

In conclusion, M. Ozan Sungurlu is honored as one of the pioneers of geology in Türkiye, who dedicated his life to his country and the study of geological sciences and left an example of a noble and fruitful life worthy of international recognition. He was an outstanding field geologist and a true patriot who always worked to improve the socio-economic conditions of the country. His example will continue to inspire and encourage new generations for many years to come to face the scientific challenges ahead and continue to explore the geology of Türkiye.

**Acknowledgments**

The author would like to thank the entire family of M. Ozan Sungurlu for their permission for publication and the Turkish Petroleum Corporation (TPAO) for their support. I would like to express my special appreciation and thanks to Kristian Schlegel, as Topical Editor for editing. I would like to acknowledge the anonymous referee for their useful remarks and positive
comments. I am appreciative of Mr. Özer Balkaş's insightful comments, edits, and suggestions regarding the text. I also thank Prof. Doğan Perinçek, Prof Yücel Yılmaz, and Remzi Aksu, PhD, for their insightful annotations and remarks on the manuscript. Additionally, I express my gratitude to Fatih Köroğlu, PhD, for his guidance. I appreciate Osman Merey and Ümmü Gülsüm Kurt's assistance.

**Data availability.** No data sets were used in this article.

**Author contributions**. This paper was conceptualized by OM. The investigation, writing, and editing were performed by OM.

**Competing interests.** The contact author has declared that neither they nor their co-authors have any competing interests.

**Funding.** No specific grants were received from public, commercial, or non-profit organizations for this research.

**M. Ozan Sungurlu's Selected Bibliography from TPAO Archive: His Contributions to Field Geology in the Türkiye**

Sungurlu MO, Soytürk N (1970). Sivas havzası ve civarının jeolojik etüdü: TPAO Arama grubu rapor no 482.

Gümüş Ö, Sungurlu MO (1970). Antalya civarında yapılan jeolojik etüt hakkında not: TPAO Arama grubu rapor no 644.

Sungurlu MO (1971). İstanbul boğazı-Bulgaristan sınırı arasında Karadeniz kıyı olanaklarının jeolojisi: TPAO Arama grubu rapor no 535.

Sungurlu MO (1972). Teke yarımadası doğu kısmının jeolojisi: TPAO Arama grubu rapor no 712.

Sungurlu MO (1973). VI. Bölge Gölbaşı-Gerger arasındaki sahanın jeolojisi. TPAO Arama grubu rapor no 802.

Sungurlu MO (1974). VI. Bölge kuzey sahalarının jeolojisi ve petrol imkanları: TPAO Arama grubu rapor no 871.

Sungurlu MO (1975). Sinop sahalarının petrol imkanları: TPAO Arama grubu rapor no 908.

Letouzey, J., Biju-Duval, B., Dorkel, A., Gonnard, R., Kristchev, K., Montadert, L. and Sungurlu, MO (1977). The Black Sea: a marginal basin. Geophysical and geological data. In: B. Biju-Duval and L. Montadert, (Editors), Structural History of the Mediterranean Basins. Editions Technip, Paris, pp. 363-376.

Sungurlu MO (1978). Doğu Toroslarda yapılacak saha çalışmaları için öneriler: Arama grubu rapor no 1225.

Sungurlu MO, Açıkbaş D, Perinçek D, Balkaş Ö (1978). Körkandil ölçülmüş stratigrafik kesiti: TPAO Arama arşiv no 7880.

Sungurlu MO, Arpat E (1978). Türkiye doğu kesiminin jeolojisi ve beklenir kabuk yapısı. TPAO Arama grubu rapor no 1204.

Açıkbaş D, Sungurlu MO, Akgül A, Erdoğan T (1979). Geology and petroleum possibilities of Southeast Turkey. TPAO Arama grubu rapor no 1410.

Yılmaz Y, Sungurlu MO, Perinçek D (1979). Cilo dağlarında eski bir okyanus kabuğu? Türkiye Jeoloji Kurumu, Altınlı sempozyumu, özel sayı, s, 45.

Şengör AMC, Yılmaz Y, Sungurlu MO (1984). Tectonics of the Mediterranean Cimmerides: nature and evolution of the western termination of Palaeo-Tethys. Geological Society, London, Special Publications, 17, 112 - 77.

Sungurlu MO, Perinçek D, Kurt G, Tuna E, Dülger S, Çelkdemir E. Naz H (1985). Elazığ-Hazar-Palu alanının jeolojisi. Petrol
İşleri Genel Müdürlüğü Bülteni, no 29, s. 83-189.

Ketin İ, Sungurlu MO (1992). GDA'da kenar kıvrımları kuşağı içinde yer alan Çermik-Kevan antiklinalinin jeolojisi hakkında tarihsel bir inceleme: TPJD Bülteni,4,1, s, 1-8.

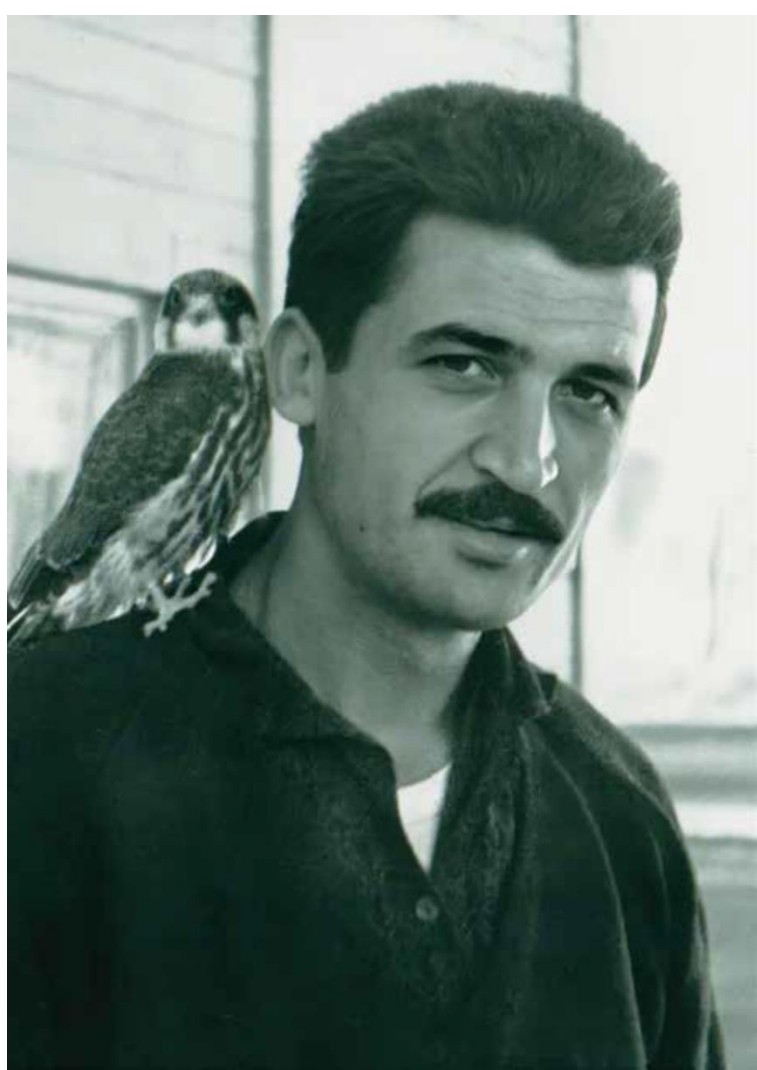

Figure 1: A photo from fieldwork 70's in Adıyaman SE Anatolia (from Dervişoğlu, 2022)

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
