# Peer review of "Mehmet Ozan Sungurlu, the Legendary Turkish Petroleum Geologist"

_History of Geo- and Space Sciences, 2023_

## Referee Comment (RC2)

General comment: a fine tribute that would benefit from the editorial input of a native speaker of English.

[referee-annotated manuscript omitted]

---

## Author Response (AR1)

**Dear Anonymous Reviewer 1**,

Thank you for your valuable comments for manuscript.

This is an interesting paper concerning a petroleum geologist, Ozan Sungurlu, who is probably poorly known outside of Turkey. The paper is an interesting biography of the personality and background life of Ozan Sungurlu, but lacks information on his scientific achievements. This should be rectified. For example, what scientific challenges did he overcome in his mapping studies? What geological observations or discoveries did he make that were previously lacking? What oil and gas fields did he contribute to the discovery of? He was clearly a pleasant and talented geologist, but the paper needs to explain better why he is held in such high regard with respect to his work as a scientist.

Author's response: I have thoroughly revised the manuscript in accordance with your comments.

The paper needs to be improved in terms of its English. Assistance from a native English speaker would be beneficial.

Author's response: I appreciate this advice. I have thoroughly revised the text with the help of a native speaker.

**Dear Anonymous Reviewer 2,**

Thank you for your valuable comments for manuscript.

Delete the word 'of '

Author response: I deleted in text.

Spell out acronym

Author response: I corrected in text.

Delete the word 'however'

Author response: I deleted in text.

A judgement call

Author response: I corrected in text.

A judgement call

Author response: I corrected in text.

Move this comment to fit chronologically , perhaps one paragraphy earlier

Author response: I corrected in text.

**Dear Prof. Yucel Yıimaz,**

I am delighted to have had the opportunity to peruse Mr. O. Sungurlu's bibliography, which was submitted to HGSS as a tribute.

He was a close friend. We collaborated on many geology projects across various parts of Anatolia. Therefore, I know him well as a friend and a competent field geologist.

After careful consideration, I confidently state that O. Mulayim's bibliography on O. Sungurlu is a well-balanced optimization that effectively reflects his geologist expertise and personal qualities.

Therefore, the publication of Mr. O. Sungurlu in its current form would be welcomed by the geological community in Turkey and worldwide.

Thank you for your valuable comments for manuscript.

I didn't attempt to improve the linguistic quality of the text as it is not my area of interest.

Author's response: I appreciate this advice. I have thoroughly revised the text with the help of a native speaker.

**Dear Dr. Schlegel,**

Thank you for the opportunity to submit a revised draft of the manuscript.

Line 6: Please give a date for the 100th anniversary.

Author response: I corrected in text as October 29, 2023

Line 9: here TPAO is mentioned for the first time, and therefore it should be fully named. In the following, the full name is not necessary (lines 39, 77).

Author response: Thank you for pointing this out. I corrected in text as Turkish Petroleum Corporation (TPAO)

Line 29: the sentence is not complete,

Author response: I corrected in text.

Line 48: Where is the Tethyan region?

Author response: I explained here. Turkey is located in an area where the Eurasian and African plates collided. As a result of this collision, not only the European and African plates merged but also small continental fragments. These fragments, from north to south, are: 1) Rhodope-Pontides fragments; 2) Sakarya continent; 3) Anatolide-Tauride platform; and 4) Bitlis massifs. The southern boundary is marked by an ophiolitic belt. The boundary between the Rhodope-Pontides fragments, the Sakarya continent, and the Anatolide-Tauride platform is marked by the ophiolitic mélange of the İzmir-Ankara-Erzincan zone, which is a suture zone of the northern branch of the Neo-Tethyan Ocean. The boundary between the Anatolide-Tauride Platform and the Arabian Plate is also marked by another ophiolitic imbricate zone, which is a suture zone formed as a result of the collision of the Anatolide-Tauride Platform and the Arabian Plate (Şengör and Yilmaz, 1981; Yılmaz, 1993). Within this framework, Turkey has seven onshore and four offshore basins of Tethyan. Most of these basins are associated with and located between and above the major suture zones of the Mesozoic and Tertiary realms of the Tethyan system. [3] The basins associated with the Tethyan realm are: 1) Southeast Anatolian Basin; 2) Tauride Platform Area; 3) Interior Basin; 4) Pontide Basin; and 5) East Anatolian Basin.

For that reason, I used the Tethyan region.

Line 49: Which universities?

Author response: I corrected in text as Istanbul Technical University and Istanbul University

Lines 60–62: There are several words repeated; the sentences are somewhat scrambled.

Author response: I corrected in text. I deleted.

Line 69: Please explain northern territories",

Author response: I corrected in text as Adıyaman and Diyarbakır territories

Line 78: Where is this well located?

Author response: I corrected in text as Adıyaman, SE Türkiye

Line 83: What means TPJD?

Author response: I corrected in text as Turkish Association of Petroleum Geologists (TAPG)

Line 88: to whom refers „his"?

Author response: I refer to Ozan Sungurlu.

Line 104: Please expand on what you mean by 'patriot".

Author response: The word "patriot" by definition means "one who loves and supports his country," according to the Merriam-Webster dictionary. I feel that the term "patriot" carries with it a higher level of pride and respect and puts a person on par with the founders of this country, who are considered the original patriots. I have used the word "patriot," which has long instilled a sense of pride in Americans. I have used it with the same meaning as Americans.